# Comprehensive Environmental Assessment of Rainwater Harvesting Systems: A Literature Review

Andréa Teston, Taylana Piccinini Scolaro *, Jéssica Kuntz Maykot and Enedir Ghisi

Laboratory of Energy Efficiency in Buildings, Department of Civil Engineering, Federal University of Santa Catarina, Florianópolis 88040-970, Brazil
* Correspondence: taylanaps@gmail.com

**Abstract:** The feasibility of installing rainwater harvesting systems in buildings is usually defined based primarily on economic analysis. In this perspective, we reviewed the literature related to water consumption in buildings, rainwater use, and environmental assessment tools to evaluate the impact of rainwater harvesting on the environment. Identifying water end uses in buildings showed a high potential for potable water savings through alternative sources (e.g., rainwater use for non-potable purposes). Most studies reviewed found potential for potable water savings from 20 to 65%. Moreover, the literature reported that rainwater harvesting systems might reduce the runoff volume from 13 to 91%. However, other possible benefits and impacts of the systems on water flow and the environment must be assessed in addition to the potential for rainwater harvesting. Life cycle assessment, life cycle cost assessment, and water balance modelling have been used in urban water management. Most life cycle studies reported that rainwater harvesting systems have better environmental performance than centralised systems. The water balance method may effectively determine the impacts these systems cause on the water cycle. Using life cycle assessment and the water balance method together is essential to evaluating rainwater harvesting systems integrated into the urban environment.

**Keywords:** urban water management; rainwater harvesting; life cycle assessment; water balance modelling

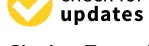



## 1. Introduction

For approximately two centuries, urban water management pursued the principles of command and control, properly represented by urban drainage, which is used for flood prevention [1]. Due to the continuous urbanisation process, drainage works have become increasingly necessary and progressively larger [2]. Thus, they have interfered more in the natural water cycle. Increasing superficial runoff, anticipating waterflow peaks, reducing evapotranspiration and the groundwater supply, and the deterioration of superficial water quality are the main urbanisation impacts on the natural water cycle [3].

Water scarcity motivates not only local studies [4–6] but also global research [7–10]. According to Kummu et al. [10], while water consumption quadrupled in the 20th century, the population suffering from water shortages increased from 14 to 58% of the world's population between 1900 and 2000. Vörösmarty et al. [9] pointed out that almost 80% of the world's population is exposed to high levels of threats to water security.

The high percentage of treated water losses in cities is also disturbing. Worldwide, the treated water loss is about 126 billion cubic metres per year, or about 30% of the water volume available for distribution [11]. Additionally, for potable water losses, such a percentage represents a tremendous waste of energy. According to Gomes [12], the energy consumption of water and sewage service providers represents about 3% of the total world electricity consumption.

The necessity to seek the balance of the hydrological cycle in cities and, consequently, to change the current model of urban water and sanitation services [13], such as hybridisation using alternative sources, is noteworthy. The water industry has been looking for alternatives for sustainable planning of the services provided [14,15]. Among the alternatives are the increase in and maintenance of permeable areas, the use of decentralised water systems, as well as changes in measurements and billing methods for services and changes in loss control and demand management [14,16,17]. Planning is therefore based on rainwater, supply, water demand for consumption, and generated sewage management.

According to Castro-Fresno et al. [18], the main objective of techniques for sustainable water management is decreasing superficial runoff by using storage tanks [2,19–29] or increasing water infiltration into the soil [30–35]. Water demand management may promote decreased water consumption, ensuring the quantity and quality required and reducing losses. Thus, it is essential to distinguish the end uses of water and the factors which influence its consumption, such as demography, user consumption, the seasons of the year, and technological devices [14].

Regarding water and sanitation supply management, there are discussions on whether the provision of sanitation services should be centralised or not [36,37]. However, centralised urban water supply systems would hardly meet the water demand due to urban population growth and climate change [38]. Because of uncertainties related to centralised systems, the number of studies in hybrid urban sanitation systems, combining centralised and decentralised systems, has increased. In hybrid systems, buildings actively participate, providing water through rainwater harvesting. In this context, evaluating the impact of such systems on the urban water balance and energy and material consumption while considering the entire life cycle of the installation is necessary.

Often, implementing or not implementing a rainwater harvesting system is defined based on economic analyses, which generally do not consider the environmental impacts generated by such systems. As a background for more accurate assessments of rainwater harvesting systems' impact in urban areas, this paper aims to analyse how the impacts of rainwater harvesting systems have been quantified. In order to support the analysis, the following topics were explored: (1) water consumption and its possible patterns in buildings, (2) rainwater usage, and (3) environmental impact studies of rainwater harvesting systems through LCA and water balance modelling.

## 2. Method

This research considers the hypothesis that rainwater harvesting systems are more environmentally sustainable than conventional water systems. Thus, the literature review was based on a critical question to be investigated: how are the impacts of rainwater harvesting systems in buildings quantified?

Thus, this work was divided into some stages. Initially, data on water consumption in buildings and their use patterns were sought. Next, the following were verified: the potential for potable water savings that can be achieved using alternative sources for non-potable purposes, how rainwater quality is addressed in research, and the impacts caused by rainwater harvesting systems in urban drainage. Finally, the environmental impact assessment tools used in the studies were investigated.

CAPES Periodicals Portal—a Brazilian platform that provides textual and reference databases in all areas of knowledge—was used as a search tool in this research. The platform makes Brazilian and international scientific content available (abstracts, articles, master's dissertations, doctoral theses, audio-visual materials, statistics, etc.). This review included scientific articles, dissertations, and theses. The results of the Brazilian dissertations and theses were included, since several works with significant quality were produced in Brazil and published only in these formats.

### 3. Water Consumption in Buildings

*3.1. Demand Management*

Demand management is fundamental to reducing potable water and energy consumption, and it is carried out through a set of instruments to optimise water use at different consumption points. Among these instruments are control, equipment maintenance, loss and leakage reduction, and awareness and education campaigns.

Ghisi et al. [39] investigated the feasibility of different strategies to save water in buildings. The strategies evaluated were greywater reuse, rainwater use, installation of water-saving equipment, and some combinations between them. Six indicators were used: potential for potable water savings, potable water savings by embedded energy or the total energy consumption index, the net present value, the internal rate of return, and the index between potable water savings and initial costs. The results showed that installing water-saving equipment to control consumption was the most viable strategy among all indicators. Beal and Stewart [40] analysed 18 months of water consumption data obtained from smart meters installed in 230 homes in Queensland, Australia. Through comparisons with historical consumption data, the authors verified that using more efficient technologies (such as water-saving equipment) and growing awareness related to water conservation reduced the degree and frequency of consumption peaks.

According to Kiperstok and Kiperstok [41], monitoring and controlling consumption in buildings are the most important actions to be taken. They should precede other investments, such as rainwater harvesting and greywater reuse. For Beal et al. [42], determining the water end uses is the first step towards obtaining relevant and successful public policies to control consumption. In order to obtain consumption profiles and water end use data, some researchers have used the smart meter system. This system comprises two elements: the meters that capture information about water use and the communication system that transmits the data in real time [16]. Statistical analysis programmes disaggregate water flow data from smart meters into different end use water categories [43,44].

Stewart et al. [16] investigated the smart metering potential for the future of water planning and management in Australian cities by providing data about water use patterns. A web-based knowledge management system (WBKMS) has been proposed to translate such data into useful information. This system integrates smart metering, water end use data, wireless communication networks, and information management systems to provide real-time data on how, when, and where water is consumed. Such data may be helpful for the consumer and water utility.

The literature indicates that when customers receive water consumption data, especially from smart meters, they are more likely to take measures that lead to water conservation. The influence of human behaviour on water conservation is a complex issue. Water user behaviour can be affected by personal factors, such as education and opinion about environmental conservation, or economic factors, such as the price associated with the water consumed, which is derived from water meter readings [45]. Furthermore, according to Howel et al. [46], smart water networks are promoted to offer leakage reduction, energy savings, water quality assurance, improved customer experience, operational optimisation, etc. For the authors, the convergence of building information modelling with smart water network models provides an opportunity to transcend existing operational barriers.

One can conclude that using smart meters can provide environmental and social benefits and help research development. Using such meters makes it possible to evaluate the performance of water-saving equipment, compare personal characteristics with water uses, and define the end uses in different types of buildings. The definition of the water end uses is valuable in terms of demand management, as it allows quantifying water demand for potable and non-potable purposes, allowing the efficient application of alternative sources.

*3.2. Consumption Profiles*

Butler and Memon [47] showed that in some countries, the average daily household per capita consumption exceeds 200 litres (USA, Saudi Arabia, and Japan). On the other

hand, in other countries, such an average is lower than 50 litres (Afghanistan, Nigeria, and Gambia), the minimum necessary to meet basic needs for hygiene and consumption, according to the United Nations World Organization [48]. In Brazil, according to the National Sanitation Information System (SNIS), the average daily per capita consumption is 152.1 L/inhabitant/day [49]. According to the National Water Agency (ANA), disregarding liquid evaporation from reservoirs, 50% of the water use in the country is intended for irrigation, 27% for urban and rural supply, 9% for industry, 8% for animal consumption, 5% for thermoelectric plants, and 2% for mining. The ANA [50] estimated a 42% increase in water withdrawals by 2040, with an increase of 26 trillion litres per year extracted from springs. Such data reinforce the need for planning actions so that uses are developed with water security, avoiding water crises and providing multiple uses of water, especially when considering the effects of climate change on the water cycle [50].

In a comparative analysis, Roshan and Kumar [51] reviewed studies carried out in 16 countries (developed and developing) from three continents: Asia, Europe, and Oceania. The average water consumption in households reported by the studies reviewed showed high variation (average of 158.8 L/person/day with a standard deviation of 51.8 L/person/day), a minimum of 60.9 L/person/day (Vietnam), and a maximum of 278.0 L/person/day (United Arab Emirates).

Aside from the differences caused by regional variations, such as climate, water availability, quality of services provided, and culture, the user's economic pattern also influences consumption [52,53]. Hussien et al. [54] investigated the influence of the economic pattern on water consumption through a case study in Dohuk, Iraq. Low-income families showed a daily per capita consumption of 241 litres, followed by 272 and 290 litres in the middle- and high-income families, respectively. In general, research indicates that the higher the social class, the more efficient the water appliances installed in the dwellings but also the higher the total water consumption [40,54,55]. These results demonstrate the importance of efficient consumption management with more demand details.

It is important to note that the charging systems adopted by water and sewage companies which adopt a minimum flat rate are an obstacle to implementing actions for the rational use of water in buildings for the low-income population. This happens because the economic aspects are essential for this part of the population, and the minimum flat rate is a disincentive to efficient water use [52].

The user age, the number of inhabitants per household, the housing type, and the season also influence daily per capita consumption [47]. The higher the number of residents per household, the lower the water consumption per inhabitant [47,54]. According to the Parliamentary Office of Science & Technology (POST) [56], per capita consumption in a single-resident household is 40% higher than the per capita consumption in a two-resident household, which is 73% higher than that in a household with four inhabitants. The research by Rathnayaka et al. [57] showed that the per capita water consumption for a person living alone is 60% higher than that for a person living with a family of six.

Dias et al. [58] analysed the socioeconomic variables of users and the variables of residential buildings in southern Brazil to identify the factors that influence water consumption. Data were collected from 89 buildings, including 3171 flats. The results showed that the per capita consumption increased with the distance from the city centre, the property age, and the land and property taxes. The per capita consumption also increased when there were sewage systems or swimming pools. Water consumption decreased with the increase in the percentage of tenants (non-owners), the average number of residents per flat, the presence of metering for each flat, and the presence of an alternative water supply system.

Several studies have used statistical analyses to associate daily per capita consumption with the characteristics of housing type, season, number of inhabitants, and climate [54,57,59–62]. Hussien et al. [54] found significant coefficients of determination between some factors and water consumption per inhabitant, such as the number of people per building ($R^2 = 0.87$), building floorplan area ($R^2 = 0.94$), number of rooms ($R^2 = 0.96$),

and garden area ($R^2$ = 0.77). The authors proposed models to predict water demand in future scenarios through such correlations.

According to Kiperstok and Kiperstok [41], the premises for rational water use in commercial and institutional buildings require a different approach from that used in residential buildings. The main reasons are (1) consumers do not pay directly for water use, (2) there is a wide variation in the habits and environmental awareness of water consumers, (3) there is not enough equipment maintenance, (4) the installations and pumping systems are more complex, making it difficult to identify leaks, (5) sectored measurement is not used, as a single measurement device is responsible for a large and complex system, and (6) water appliances are used more than in residential buildings and tend to fail more frequently.

Table 1 shows the results of water consumption in educational institutions in Brazil. The variation in daily per capita consumption between the results found by the studies is up to 37.3 L/person/day. Based on Ywashima [63], it is possible to observe a 54.4% water consumption reduction after the installation of water-saving appliances.

**Table 1.** Water consumption profiles in educational institutions in Brazil.

| Building Typology | Reference | Floor-Plan Area (m$^2$) | Number of Occupants (Person) | Daily per Capita Consumption (L/Person/Day) | Daily Consumption per Area (L/m$^2$/Day) | Total Daily Consumption (L/Day) |
|---|---|---|---|---|---|---|
| Higher education institution | Almeida [64] | 2639.70 | 220 | 19.7 | 1.64 | 4332 |
| | Marinoski and Ghisi [65] | 5149.45 | 565 | 15.5 | 1.70 | 8758 |
| Full-time schools and kindergartens | Ywashima [63] (Before [3]) | - | 111 | 43.0 | - | 4773 |
| | Ywashima [63] (After [3]) | - | 111 | 19.6 | - | 2172 |
| Part-time schools | Ywashima [63] (Before [3]) | 2010.84 | 577 [1] | 23.6 | 6.77 | 13,617 |
| | Ywashima [63] (After [3]) | 2010.84 | 577 [1] | 10.7 | 3.08 | 6196 |
| | Fasola et al. [66] | 638.00 [2] | 149 | 5.7 | 1.33 | 849 |
| | Fasola et al. [66] | 800.00 [2] | 253 | 7.4 | 2.34 | 1872 |

[1] Average number of students in 2002 (564 students), 2003 (585 students), and 2004 (581 students). [2] Floorplan area was considered equal to the roof area. [3] Before and after the installation of water-saving appliances.

Nunes [67] and Gois et al. [68] evaluated water consumption in shopping centres located in two Brazilian cities: Rio de Janeiro and Londrina, respectively. In both studies, the buildings have a floorplan area of 135,000.00 m$^2$, and the daily consumption per inhabitant was evaluated considering the fixed and mobile population. The shopping centre investigated by Nunes [67] showed a daily per capita consumption of 18.92 L/person/day. The shopping centre located in Londrina showed lower consumption (14.87 L/person/day) [68,69]. This difference can be explained based on the climate, number of users, and number of water appliances.

Companies that wash clothes and vehicles are significant water consumers among the service providers. According to Almeida et al. [70], bus-washing stations play an important role in the daily life of an urban metropolis. In São Paulo, the most populous city in Brazil, 15,064 vehicles circulate on weekdays and are washed at the end of each day, consuming approximately 2,200,000 m$^3$ of water per year.

Some results of Brazilian and foreign research are shown in Table 2. There is a high disparity in the results. In the Brazilian studies reviewed, consumption for washing cars varied between 70 and 250 L/vehicle/day. The water demand for this service also depends on many variables, such as the technology used, making it difficult to characterise a consumption profile.

**Table 2.** Water consumption for washing vehicles in different countries.

| Reference | Service Type | Location | Average Water Consumption (L/Vehicle) |
|---|---|---|---|
| Brown [71] | Car wash (self-service) | EUA | 45 |
| | Car wash (tunnel) | EUA | 268 |
| Al-odwani et al. [72] | Car wash | Kuwait | 185–370 |
| Zaneti et al. [73] | Car wash | Porto Alegre, Brazil | 115–119 |
| Morelli [74] | Car wash | São Paulo, Brazil | 150–200 |
| | Bus wash | São Paulo, Brazil | 400–600 |
| Ghisi et al. [75] | Car wash | Brasília, Brazil | 150–250 |
| Lage [76] | Car wash | Belo Horizonte, Brazil | 95 |
| | Car wash | Belo Horizonte, Brazil | 70 |

The hotel sector also showed a high variation in the water consumption profile. Barberán et al. [77] evaluated the water savings in a hotel in Zaragoza, Spain after replacing the water appliances with water-saving technologies. The results showed savings of 21.4% (i.e., the consumption decreased from 321 L/guest/day to 252 L/guest/day). The literature review conducted by Gössling et al. [78] reported consumption ranging from 90 L/guest/day to 1596 L/guest/day. The consumption profiles in hotels depend on the end uses of water, which can vary depending on the services and amenities offered. According to Tortella and Tirado [79], all-inclusive hotels have higher water consumption due to meal consumption and the higher use of facilities and services with high water consumption intensities. In Brazil, according to Nascimento and Sant'ana [80], the water consumption in two analysed hotels was 2.08 L/m$^2$/day for Hotel A and 4.44 L/m$^2$/day for Hotel B. According to the authors, this difference was due to the higher diversity in infrastructure at Hotel B.

Table 3 shows the results of daily water consumption found by Proença and Ghisi [81] and Nunes [67] for office buildings in Brazilian cities. The per capita water consumption varied between 34.9 and 101.6 L/person/day. According to Proença and Ghisi [81], the Aliança and Granemann buildings presented higher water consumption, probably due to the great diversity of uses (Aliança) and the significant floating population (Granemann). Despite the wide variety of results for daily consumption per person found in the buildings, the average of 58.5 L/person/day is similar to the one estimated by the water and sewage service in the state of São Paulo, Brazil for office buildings (i.e., between 30 and 50 L/person/day) [82].

Kammers and Ghisi [83] carried out a study in 10 public buildings located in Florianópolis in southern Brazil. The results showed variation in the daily per capita consumption from 28.00 to 67.20 L/person/day (Table 4). According to the authors, this was due to the difference of the water appliances in the buildings. The largest consumers, Celesc and the Secretary of Agriculture, have a cooling tower for air-conditioning. There was no significant variation between the other buildings (i.e., an average of 30.29 L/person/day and a standard deviation of 6.07 L/person/day).

Through the studies on water consumption profiles reviewed in this section, it was possible to observe a high variation in results due to several factors influencing consumption, such as leaks, floating population, and the time users stayed in the building. Therefore, these data are helpful for case studies and should only be used in other studies after in-depth analysis.

**Table 3.** Water consumption profiles in office buildings.

| Reference | Building Type or Name | Location | Total Number of Users (People) | Daily per Capita Consumption (L/People/Day) |
|---|---|---|---|---|
| Nunes [67] | Commercial | Rio de Janeiro, Brazil | 10,000 | 54.6 |
| Proença and Ghisi [81] | Aliança | Florianópolis, Brazil | 157 | 84.1 |
| | Exaldo Moritz | Florianópolis, Brazil | 96 | 65.4 |
| | Granemann | Florianópolis, Brazil | 51 | 101.6 |
| | Ilha de Santorini | Florianópolis, Brazil | 148 | 53.7 |
| | Ilha dos Ventos | Florianópolis, Brazil | 76 | 34.9 |
| | Manhattan | Florianópolis, Brazil | 138 | 39.7 |
| | Olmiro Faraco | Florianópolis, Brazil | 143 | 48.6 |
| | Pedro Xavier | Florianópolis, Brazil | 243 | 51.9 |
| | Trajanus | Florianópolis, Brazil | 128 | 55.4 |
| | Via Venneto | Florianópolis, Brazil | 100 | 53.6 |

**Table 4.** Water consumption profiles in public buildings in Brazil. Based on Kammers and Ghisi [83].

| Building Type | Floorplan Area (m²) | Number of Occupants (Person) | Occupant Area Ratio (m²/Person) | Daily per Capita Consumption (L/Person/Day) | Daily Consumption per Area (L/m²/Day) | Total Daily Consumption (L/Day) |
|---|---|---|---|---|---|---|
| Badesc (development agency) | 1300.00 | 165 | 7.88 | 29.00 | 3.68 | 4785.00 |
| Celesc (electricity distribution company) | 21,405.00 | 1035 | 20.68 | 67.20 | 3.25 | 69,552.00 |
| Crea (regional council of engineering and agronomy) | 2000.00 | 95 | 21.05 | 32.90 | 1.56 | 3125.50 |
| Deter (department of transport) | 1400.00 | 107 | 13.08 | 31.50 | 2.41 | 3370.50 |
| Epagri (agricultural research and rural extension company) | 8025.00 | 324 | 24.77 | 29.70 | 1.20 | 9622.80 |
| Secretary of Agriculture | 3726.00 | 197 | 18.91 | 57.30 | 3.03 | 11,288.10 |
| Secretary of Education | 6800.00 | 520 | 13.08 | 18.30 | 1.40 | 9516.00 |
| Secretary of Public Safety | 1690.00 | 90 | 18.78 | 33.10 | 1.76 | 2979.00 |
| Audit office | 8200.00 | 542 | 15.13 | 28.00 | 1.85 | 15,176.00 |
| Court of justice | 13,617.00 | 1216 | 11.20 | 39.80 | 3.55 | 48,396.80 |

### *3.3. Water End Uses*

The water end uses can be assessed through questionnaires and interviews [54] (the results found in the interviews are analysed together with the total water consumption in the building and flow of water appliances) [81,83–85] and through real-time smart measurement systems [40,86,87].

Possible failures should be considered in research carried out only through questionnaires and interviews, as the perception of users regarding their water use can be imprecise [40,88]. Characterising end uses through measurement and sociodemographic information helps identify correlations between the population groups' behaviour and their perceptions of water use. According to Beal et al. [40], the characteristics of the people group who overestimate their water consumption have lower incomes and levels of education, lower numbers of children in the family, lower use of efficient technologies, and generally small domestic occupations. On the other hand, the people who underestimate their consumption have higher family incomes, a higher number of young children, and higher use of efficient technologies, such as showers with lower flow and washing machines classified as having lower consumption. Table 5 shows some case studies on the contribution of water end use to domestic consumption in different countries.

**Table 5.** Case studies on water end uses in residential buildings in different countries.

| End Uses | Beal et al. [42] | Matos et al. [89] | POST [56] | Hussien et al. [54] | Jiang et al. [90] | Sebusang and Basupi [91] | Average | Standard Deviation |
|---|---|---|---|---|---|---|---|---|
| | Australia | Portugal | UK | Iraq | China | Botswana | | |
| Shower (%) | 29.5 | 6.0 | 5.0 | 14.0 | 21.7 | 27.1 | 17.9 | 9.7 |
| Washbasin (%) | - | 34.0 | 9.0 | 32.0 | - | 9.4 | 21.1 | 13.8 |
| Cleaning (%) | - | - | - | 5.0 | 8.8 | - | 6.9 | 2.7 |
| Kitchen tap (%) | 19.0 | 32.0 | 15.0 | 19.0 | 16.0 | 17.7 | 19.8 | 6.2 |
| Toilet (%) | 16.5 | 23.0 | 31.0 | 9.0 | 14.0 | 20.1 | 18.9 | 7.6 |
| Washing machine (%) | 21.0 | 2.0 | 20.0 | 13.0 | 31.4 | 2.8 | 15.0 | 11.4 |
| Bathtub (%) | 1.0 | - | 15.0 | 0.0 | - | 14.3 | 7.6 | 8.2 |
| External use (%) | 5.0 | - | 4.0 | 80 | 1.2 | - | 4.6 | 2.8 |
| Losses (%) | 6.0 | - | - | - | - | - | 6.0 | 0.0 |
| Dishwasher (%) | 2.0 | 3.0 | 1.0 | - | - | 0.0 | 1.5 | 1.3 |
| Drinking (%) | - | - | - | - | 6.9 | - | 6.9 | 0.0 |
| Laundry tap (%) | - | - | - | - | - | 1.9 | 1.9 | 0.0 |
| Kitchen (other) (%) | - | - | - | - | - | 2.3 | 2.3 | 0.0 |
| Bath (others) (%) | - | - | - | - | - | 4.3 | 4.3 | 0.0 |

There is a significant difference between the water end uses in the studies addressed due to climate, socioeconomic class, domestic appliances, culture, etc. The results of Sebusang and Basupi [91] presented in Table 5 are the end uses for all groups analysed (low-, medium-, and high-income). The authors also presented the water end uses of these groups separately. The results showed that although end uses varied according to income, for all groups considered, the water end use for toilets was around 18.5–24.4%, suggesting that it is not highly affected by income. This information is essential, since the toilet is one of the main appliances where non-potable water can be used.

According to the end uses in the studies reviewed by Roshan and Kumar [51], European countries use less water to wash dishes and clothes than Asian countries. Among European countries, the average water consumption for washing dishes was 9.9 L/person/day (standard deviation of 14.6), and for washing clothes, it was 17.4 L/person/day (standard deviation of 7.2). In Asia, the average (standard deviation) was 23.8 L/person/day (13.6) and 26.3 L/person/day (14.2) for washing dishes and clothes, respectively. The possible reasons for these differences are access to water-saving appliances (such as automated washing machines and mechanised dishwashers), the availability of water-efficient technologies, and environmental awareness. Other water end uses such as toilet flushing, showers and

bathtubs, indoor taps, and leakages and others were also reported. All European countries for which end use data were reviewed are developed countries. On the other hand, considering water end use data from Asian countries, data from developed and developing countries were collected. Thus, the authors emphasised that the water end uses from different regions should be compared with caution and while considering all factors that affect consumption. These factors include annual climate variation, economic and social development, scientific and technological advances, and differences in people's behaviour.

In Brazil, a compilation of research on the water end use in residential, public (public administration and educational institutions), and commercial buildings was presented by Teston et al. [92]. Table 6 shows the average end uses found among the studies reviewed by the authors for each building typology.

**Table 6.** Water end uses of case studies in residential, public, and commercial buildings. Based on Teston et al. [92].

| End Uses | Residential Buildings | Public Buildings | Commercial Buildings |
|---|---|---|---|
| | Average | Average | Average |
| Toilet (%) | 22.1 | 44.1 | 67.6 |
| Washbasin (%) | 8.2 | 15.4 | 25.4 |
| Bath (%) | 25.2 | - | - |
| Urinal (%) | - | 17.1 | - |
| Kitchen tap (%) | 16.9 | 26.5 | - |
| Drinking fountain (%) | - | 1.9 | - |
| Laundry sink or washing machine (%) | 17.0 | - | - |
| Cleanings (%) | 2.4 | - | 2.3 |
| Car wash (%) | - | 2.0 | - |
| Cooling tower (%) | - | 22.7 | - |
| Others (residential) [1] (%) | 12.9 | - | - |
| Others (public) [2] (%) | - | 6.2 | - |
| Others (commercial) [3] (%) | - | - | 11.7 |

[1] Others (residential): irrigation, garage tap, laundry tap, car wash, and sidewalk washing. [2] Others (public): cleaning, irrigation, taps, and lab showers. [3] Others (commercial): water used to make coffee, wash fruits, drink, and other uses from the kitchen tap.

Thus, it was observed that the shower is the main responsible factor for household water consumption, followed by the toilet and washing clothes. Considering that the toilet, washing clothes, and the category "others (residential)" can use non-potable water, the authors reported that the potential for potable water savings in this type of building varied between 24.0% and 79.0%. Concerning end uses in public buildings, the toilet and urinal together account for more than half of water consumption (61.2%). In this case, considering the percentages related to the toilet, urinal, car washing, and "others (public)" as non-potable uses, the minimum potential for potable water savings was 49.1%, and the maximum was 82.1%. In commercial buildings, the toilet was responsible for the highest percentage of water end use, showing a high potential for potable water savings in this building typology (about 70%) [92].

Based on the case studies reviewed, the high potential for replacing potable water with lower quality water was observed for uses that did not require potability in different types of buildings. Through the analysis of the water end use in buildings, it is possible to assess the potential for potable water savings from using rainwater (for non-potable purposes) in urban buildings. Although computer programmes allow the simulation of rainwater-harvesting systems with different rainwater demands, the closer these values are to reality, the more accurate the environmental assessment of systems in the hydrological cycle.

## 4. Rainwater Harvesting

Generally, rainwater-harvesting systems comprise a catchment surface, distribution pipes, rainwater tanks, and complementary devices. In such systems, the building's roof

is usually used as a catchment surface. During rainfall, roof runoff is conducted to a storage tank and piped to water appliances through motor pumps and an independent piping system. When the tank capacity is insufficient for storing all the water collected, the overflow water is directed to the urban rainwater collection system or infiltration ditches. Complementary devices, such as first-flush diverters, solids removal filters, and fine filters, may contribute to water quality control by reducing the risks of user exposure to pathogens [93]. The rainwater-harvesting systems also contribute to net-zero water buildings as an alternative water source and can significantly and positively impact water-stressed cities [94,95].

*4.1. Potential for Potable Water Savings*

Several authors have studied the potential for potable water savings through rainwater usage [26,96–101]. The potential for potable water savings is associated with the percentage of potable water that may be replaced with rainwater (rainwater demand), according to the water end uses. Additionally, such a variable depends on the local rainfall, roof area, catchment surface type, rainwater demand, and potable water demand. Therefore, the most accurate procedure for sizing rainwater tanks is the computer simulation method, using rainfall and demand data with the minimum daily resolution [102]. Both rainfall and demand data must be representative of the study case, considering the annual variations (not considering the seasonal irrigation can introduce errors in the tank yield). Furthermore, it was observed that increases in the roof area increased the tank yield until a certain point. From this point, the yield was limited by the tank size. Similarly, increases in the tank size increased the tank yield until the tank size became so large that it stored most of the rain inflow [103].

Crosson [104] analysed a midsize office project in a severe drought in Los Angeles (CA, USA) that aimed to achieve net-zero water. In this sense, harvesting rainwater is fundamental to meeting the potable water demand. The authors investigated the role of the catchment size, storage capacity, and infiltration area in achieving performance goals. The results showed that storage size was relevant up to a limit, while the catchment area (and corresponding infiltration) was critical in this project.

Considering the size of the rainfall series used in different simulations, some researchers have evaluated the influence of the series' size on the potential for potable water savings when designing a rainwater-harvesting system [99,105]. Geraldi and Ghisi [105] compared the usage of 30-year daily rainfall data from Berlin, Germany with smaller series for simulating rainwater-harvesting systems. The results showed that 10-year daily series are sufficient for sizing rainwater tanks.

Due to the climatic data necessary for sizing rainwater-harvesting systems, it is noteworthy that climate changes may modify the predictions of rainwater usage. Wallace et al. [106] developed a methodology to simulate rainwater-harvesting systems' performance using predicted climate data. The method was tested through application in two climatic regions of the Federated States of Micronesia in the western Pacific, where most of the population depends on rainwater harvesting for their potable water supply.

The computer programme Netuno (LabEEE, Florianópolis, Brazil) has been used in Brazilian research [39,75,107,108] to size rainwater tanks by simulations of rainwater-harvesting systems. Its methodology is based on behavioural models, providing results such as the available volume in the rainwater tank before consumption, rainwater volume consumed in a day, the association between the potential for potable water savings and the rainwater tank's volume, and the overflowed rainwater. Netuno (LabEEE, Florianópolis, Brazil) also estimates the ideal rainwater tank capacity through a predetermined interval between the storage tank volumes (m$^3$) and an index of potential differences in potable water savings (%/m$^3$). Umapathi et al. [109] used real-time monitoring of the rainwater-harvesting systems in the East Queensland region of Australia. They intended to analyse the potential for water savings. The study highlights the need to correctly size tanks because

they realised that although rainwater harvesting contributed to about 31% of the total household demand, about 13% of the potable water was used to supply rainwater tanks.

Reliability and efficiency may also be evaluated to determine the systems' performance. Zhang et al. [110] calculated the rainwater-harvesting systems' reliability using the relationship between the period when the demand is met with rainwater and the total period investigated. The efficiency of rainwater harvested may be shown by the ratio between the volumes of stored and collected rainwater, considering the losses due to the catchment area characteristics [111]. Therefore, the system reliability is associated with verifying demand fulfilment, while efficiency is related to the system's impact on surface runoff due to rainwater retention.

Domènech and Saurí [98] compared the rainwater-harvesting systems in single and multi-family buildings in Barcelona's metropolitan region (Spain), assessing the social, economic, and potable water consumption impacts. All residents interviewed showed satisfaction with contributing to the environment by using rainwater. Regarding the potential for potable water savings, rainwater usage may reduce per capita water consumption in single-family buildings, meeting 100% of the demand for toilets and gardening. In the multi-family buildings investigated, rainwater was used only for gardening and represented significant potable water savings per building. Some buildings had greywater reuse systems to meet the toilet demand [98] since, in multi-family homes, the roof runoff is usually not large enough to meet such demand by means of rainwater.

Al-Houri and Al-Omari [112] showed that the rainwater-harvesting systems in northern Jordan are a technically and socially viable solution to mitigating the region's water scarcity problem. Rainwater can supply between 7.6% and 16.8% of domestic water consumption.

Cook et al. [113] evaluated the implementation of a collective rainwater system to meet the potable water demand. According to the authors, the collective use of rainwater may offer advantages such as greater cost-effectiveness, ecological footprint reduction, and quality control through centralised disinfection. The rainwater collected from roofs was sufficient to meet about 90% of the demand. However, the energy consumption associated with this water source was higher than that in centralised systems.

A compilation of Brazilian research studies about the potential for potable water savings in residential buildings was carried out by Teston et al. [102]. The authors showed that most cases (more than 55%) presented potential for potable water savings between 26.1% and 48.1%. Additionally, more than 40% of the cases showed that the systems presented reliability between 93.5% and 100%. In approximately 65% of the cases, the system's reliability exceeded 75%. Therefore, it is possible to conclude that rainwater presented promising results in meeting the demand in residential buildings.

Rainwater harvesting in homes in a region with water scarcity problems in northeastern Brazil could save up to 25% of potable water [114]. Kolavani and Kolavani [115] also assessed the potential for potable water savings by rainwater usage in the residential sector in seven cities in northern Iran. By using rainwater-harvesting systems in homes, it would be possible to achieve a potential for potable water savings between 15.5% and 19.1%. In Palmares and Caruaru, Brazil, the potential for potable water savings by using rainwater collected from roofs was 51.1% and 44.4%, respectively [116].

Some studies have evaluated the impact of rainwater harvesting on potable water consumption in educational institutions, as compiled by Teston et al. [102]. In such buildings, there was a high frequency for replacing from 69.6 to 80.6% of potable water demand with rainwater. Furthermore, the highest potential for potable water savings was 53.2%. When analysing the systems' reliability, it is noteworthy that the rainwater demand was met more than 65% of the time, with an associated frequency of 60%. However, only 11% of the data presented a reliability higher than 76%, a result obtained due to the high demand for potable water. According to the authors [102], the highest reliability obtained (87.7%) corresponded to the system with the lowest monthly rainwater consumption, while the highest consumption had reliability of 31.5%.

After analysing water consumption from 100 public schools in southern Brazil, Antunes and Ghisi [117] performed rainwater-harvesting system simulations for two institutions: one with high per capita water consumption and the other with low consumption. According to the authors, installing two 15-m$^3$ rainwater tanks in the institution with the highest water consumption could provide potable water savings of between 32.7% and 62.5%. Installing a 15-m$^3$ rainwater tank in the school with lower water consumption would provide potable water savings from 53.3 to 78.1%. The variations in the potential for water savings were due to the variation in rainwater demands considered in the study (60, 70, and 80% of the buildings' potable water consumption).

Almeida et al. [118] studied rainwater-harvesting systems combined with an extensive green roof in university buildings. It was verified that this combination reduces the volume of water saved (mainly due to the additional water retention and storage capacity of the green roof), but it increases the volume of water retained. Considering 50% of the catchment area to be covered by a green roof, the authors reported a reduction of less than 6% in the potential for water savings and an increase of about 15% in the retained water volume.

Klein [119] investigated the potential for potable water savings of different green and conventional roofs through prototypes constructed in Florianópolis in southern Brazil. It was observed that the potential for potable water savings in buildings with green roofs highly depends on the catchment area and water demand. The author found that the potential for potable water savings ranged from 8.66 to 44.99% for green roofs and from 25.72 to 46.21% for conventional roofs.

Silva and Ghisi [120] conducted a sensitivity analysis of design variables and rainwater-harvesting systems' performance, also considering different rainfall patterns. The potential for potable water savings and the rainwater tank capacity were considered dependent variables. In general, the daily potable water demand, rainwater demand, and the catchment area were the most influential variables for sizing rainwater tanks and for the potential for potable water savings. The rainwater demand was the most influential variable on the potential for potable water savings for most cities analysed.

Lopes et al. [101] analysed the potential for potable water savings in residential buildings by varying the design parameters. They concluded that (1) increasing daily rainwater demand increases potable water savings and the optimal rainwater tank capacity, (2) for small catchment areas, the relationship between the increase in daily rainwater demand and the optimal rainwater tank capacity is not linear, and (3) cities and regions with high rainfall tend to need smaller rainwater tanks, with greater potable water savings.

Ghisi et al. [75] assessed the potential for potable water savings by rainwater usage for washing vehicles in the capital of Brazil. The authors varied the catchment area and water demand, obtaining a potential for potable water savings between 9.2% and 57.2%. Another evaluation considering the same usage purpose conducted by Lage [76] concluded that with viable and attractive investments, it is possible to obtain potential for potable water savings between 9.74% and 26.8% in another Brazilian city. The difference in results may be attributed to the variation in the input data considered in the first study but not in the second, as well as the difference between the rainfall patterns considered in each study.

### 4.2. Impacts on Drainage Systems

In addition to promoting potable water savings, rainwater-harvesting systems may improve urban surface runoff management [121]. Rainwater harvesting is one of the low-impact development solutions used to restore the natural water cycle in cities [29]. Zahmatkesh et al. [27] evaluated a controlled scenario of such solutions in NY, USA. Three practices were implemented: rainwater harvesting, permeable pavements, and bioretention. According to the authors, the techniques may reduce the annual rainfall volume drained by, on average, 41%, reducing peak flows by 13% in a low-rainfall scenario, 11% in a medium-rainfall scenario, and 8% in a high-rainfall scenario.

Another technique to reduce the peak flows of surface water runoff used in urban centres is retention tanks. Such storage tanks allow slow water flow through a small outlet

hole. The retention and rainwater-harvesting tanks differ in terms of purpose. While the first must remain empty to guarantee the subsequent rainfall volume storage [22], the second must remain full to meet the demand. Researchers have proposed methods for sizing rainwater tanks that also satisfactorily decrease surface runoff peaks [28,122]. Gee and Hunt [28] assessed passive and active systems. The passive system uses a larger-than-ideal tank, with a space reserved only for rainwater storage to mitigate peak flow. The storage of collected water for consumption is performed at a certain level such that the upper part of the tank is available to store rainwater for the next rain event. Above this level, there is a passive release hole which operates in the same way as in the retention tanks studied by Tassi [2].

As an alternative to the passive release approach, there is the active release approach. Such a system includes a real-time control device which automatically releases collected water based on predicted rainfall and the water level in the storage tank. Using the National Weather Service's predicted rainfall, the device activates the system to slowly release the required water volume, ensuring that the predicted rainfall volume is stored. Water is only released if the storage capacity is insufficient to store the predicted rainwater volume [28]. The passive system achieved an average of 82% volume reduction and 90% peak reduction, and the active system achieved 91% and 93%, respectively. The enhanced system with real-time control may provide additional benefits by mitigating flooding through runoff storage and attenuating downstream flows [123]. Shetty et al. [124] demonstrated that connecting a smart retention tank (cistern) to a green roof maximises rainwater collection. The retention tank was designed to be completely empty 24 h after a storm event ended or partially empty if a subsequent storm was predicted. The total rainfall retained by the green roof over the monitoring period increased by 10% if using the retention tank.

The ability of dual-substrate layer extensive green roofs to retain rainwater compared to a conventional green roof (single-substrate layer extensive green roof) was investigated by Wang et al. [125]. The authors found that considering an adsorption layer for water retention (mixture of activated charcoal with perlite and vermiculite), aside from the nutrition layer for plant growth, increased rainfall retention (between 55.4% and 65.9% versus 52.5% for the conventional green roof). Furthermore, such an absorption layer is also recommended when aiming for pollution reduction.

Other studies have evaluated the impact of rainwater-harvesting systems on drainage. Considering a small area with some buildings, the installation of storage tanks of 10 $m^3$ every 100 $m^2$ could reduce the runoff and peak flow by 18% and 20%, respectively [24]. Palla et al. [29] found an average peak reduction rate of 33% and a reduction in rainfall volume of 26%. Zhang et al. [19] obtained a reduction in runoff volume of 13.9%, 30.2%, and 57.7% for 207.2 mm, 95.5 mm, and 50.0 mm of daily rainfall, respectively. Research performed in the USA evaluated the benefits of rainwater management by rainwater-harvesting system usage [23]. As a result, it was found that rainwater harvesting may reduce the runoff volume by up to 20% in semi-arid regions. This percentage is lower for regions with higher rainfall. A study conducted in a residential condominium of single-family buildings pointed to decreases of 4.9% and 4.4% in the peak of surface runoff for two storage tanks. The design rain considered was the highest daily rainfall of the series [102].

### 4.3. Water Quality

Domestic rainwater usage may cause damage to users' health depending on its quality. Rainwater runoff from roofs and the high concentration of pollutants in the atmosphere affect rainwater quality. Thus, harvested rainwater must meet specific water quality parameters to preserve users' health and the rainwater-harvesting system's life cycle, even for non-potable purposes. Although no internationally known standard regulates rainwater quality for non-potable purposes [126], rainwater needs treatment before being stored and consumed [127].

Gwenzi et al. [128] warned of the risks related to harvested rainwater consumption in public health. Contamination mechanisms include atmospheric deposition, leaching of

roofing materials, and faecal contamination by animals and humans. The meteorological conditions, land use practices, catchment materials, temporal patterns of hydrological factors, and their interactions are important for rainwater quality. Rainwater contamination occurs in three stages [129,130]. The first occurs when the rainfall washes gases and fine volatile particles from the urban atmosphere, the second refers to the catchment surface washing, and the third is associated with the storage conditions.

In the first stage, the source of contamination may be local road traffic and industrial activities. The rainwater pH in urban areas varies from 4.5 to 10.4, showing that anthropogenic activity may significantly affect this parameter. In the second stage, contamination by faecal bacteria is common, often being detected in all rainwater samples [129]. In addition, during the second stage, physical-chemical contamination is expected, with the possibility of heavy metal detection from wet or dry depositions on the catchment surfaces. In addition, roofing and gutter material may be a source of chemical compounds, particularly with metals from the leaching process on painted roofs, as in the case of lead-based paints. Pesticides from agricultural activities may also reduce the quality of harvested rainwater. During the storage stage, pH increase and sedimentation are two physical phenomena contributing to improving harvested rainwater quality [129].

In Australia, Sharma and Gardner [131] found that the microbiological quality of rainwater from storage tanks is unsuitable for any potable use without a prior disinfection process. They also warned that consumers should be careful about the water's chemical quality, primarily due to lead detection.

Tengan and Akoto [132] assessed the risks associated with human health due to heavy metal contamination in rainwater runoff from metal and asbestos roofing in the Republic of Ghana. This research showed that the runoff obtained was unsuitable for potable uses. The ingestion or dermal absorption of rainwater runoff from any of the roofs analysed could cause carcinogenic and non-carcinogenic health risks due to the high cadmium doses to which adults and children would be exposed. Another similar study in Palestine showed that most rainwater runoff samples from roofs presented heavy metal concentrations below the limits established by WHO for potable purposes. Additionally, health risk analyses indicated that the collected samples might be considered safe for human consumption [133].

Storage tank usage and maintenance are also important. Although the maintenance process is simple, it is often not performed or not performed correctly [134]. Such lack of maintenance causes problems in pumping, increases the risk of disease, impairs water quality maintenance by limiting its use, and causes problems in pipes and valves. Among seven houses evaluated in a Brazilian semi-arid region, only one house had no *Escherichia coli* detection in its harvested rainwater samples [135].

In order to improve the harvested rainwater quality, some strategies may be adopted, such as diverting the first flush. The trend of improvement in water quality is notable when increasing the volume of the first flush diverted [136–138]. There is wide variation regarding the initial runoff volume recommended for diversion, ranging from 0.5 mm to 2.0 mm [139]. Amin et al. [140] stated that diverting the first drained millimetre improves the microbial rainwater quality. Gikas and Tsihrintzis [141] noted that using a first-flush device improved the physicochemical rainwater quality but could not prevent its microbial contamination. Therefore, for potable purposes, the authors recommend adopting disinfection strategies.

Gikas and Tsihrintzis [142] assessed the impact of the number of antecedent dry weather days on the quality of rainwater collected from two types of roofs, considering periods from 1 to 4, 5 to 8, 9 to 12, 13 to 16, and more than 16 preceding days without rainfall. The parameters analysed were electrical conductivity, nitrite, nitrate, ammonia, total phosphorus, chloride, sulphate, magnesium, and calcium. The study indicates a tendency to reduce pollutants as the number of dry days increases for the concrete roof. The opposite behaviour was observed for the ceramic tile roof. Such results were justified by the fact that the concrete roof was horizontal, and the locally accumulated pollutants were removed by wind [142]. That aside, according to Zhang et al. [143], the roof material significantly affects the rainwater quality. Comparing concrete, asphalt, ceramic tiles, and

green roofs, the authors reported that the ceramic tile roof was most suitable for rainwater harvesting (lowest mean concentration in 17 water quality parameters). The rainwater drained from green roofs was observed to have more pollutants than natural rainwater. However, such drained water can potentially be used for non-potable purposes [144]. When comparing green roofs with concrete roof tiles, Teixeira et al. [145] found that the rainwater collected from the concrete tiles presented lower turbidity and chemical oxygen demand in most of the results. The green roof contributed to decreasing the natural acidity of the rainwater. Morales et al. [146] also showed the tendency of green roofs to neutralise water (i.e., such roofs are a viable alternative for mitigating the problem of acid rain in urban centres).

To analyse another conception of green roofs, Xu et al. [147] performed an experiment using hydroponics. They merged rainwater harvesting with greywater treatment in the experiment. The hydraulic retention time of 8 days was considered the best among those analysed in the laboratory. Therefore, the experiment was installed in a motorhome. The experiment showed that the effluent water quality was improved, with average removal rates for the chemical oxygen demand, 5-day biochemical oxygen demand, methylene blue active substance, and turbidity reaching 84%, 98%, 91%, and 86%, respectively. The effluent could be used for vessel discharge, toilets, irrigation of landscapes, and washing roads, as recommended by Teixeira et al. [145].

Assessing water usage for non-potable purposes, the presence or absence of pathogens or pollutants does not always constitute a significant risk. As in sewage effluents, to be characterised as a real risk, the pathogen in water must resist treatment processes, survive in the environment in sufficient numbers to infect an individual with whom it comes into contact, and cause disease or subsequent transmission [148].

Fewtrell et al. [149] performed a quantitative risk analysis for rainwater-harvesting systems while considering the systems' associated risks: drowning or near-drowning, injury, and infection. The first risk is associated with human access to the lower rainwater tank, the second with maintenance and cleaning, and the third with infection risk, the consumption of non-potable water (by direct ingestion or aerosolised particles from toilet flushes), or consumption of vegetables watered with stored rainwater. The rainwater-harvesting system assessed consisted of a solids removal filter and a lower rainwater tank, and rainwater was used for watering the garden and flushing toilets. According to the research, there is a greater risk related to injury due to maintenance of the gutters than microbiological contamination.

## 5. Environmental Assessment Tools

Among the studies that assess the environmental impact of rainwater-harvesting systems are life cycle assessment (LCA) and water balance modelling. The first is based on analysing all life cycle environmental impacts of a product or service from manufacturing to its final disposal. The second is related to the water cycle, namely the balance of water inflows and outflows in the urban environment.

### 5.1. Life Cycle Assessment

LCA has been used for decision making between products, services, or activities [150,151] for the global assessment of the environmental impact caused during the life span and by the assessment of the life cycle cost [152–155] and the social life cycle assessment [156–159].

Regarding environmental impact, LCA allows assessments and comparisons between environmental parameters (such as toxicity and eutrophication) or life cycle phases from the cradle to the grave [160]. According to ISO 14040 [161], LCA can support opportunities to improve the environmental performance of products in the different life cycle phases from acquiring raw materials to production, use, recycling, and final disposal. LCA studies are divided into four phases: (1) definition of the goal and scope, (2) inventory analysis, (3) assessment of life cycle impacts, and (4) interpretation.

The first phase covers the characterisation of the product or service, including its function, functional unit, reference flows, the system boundary (i.e., the processes that will be considered), the impact categories and evaluation methods, assumptions, and limitations [161]. Loubet et al. [151] reviewed LCA in urban water systems. They found that more than 50% of the studies analysed adopted the volume of 1 m³ of water per user as a functional unit, which means collecting, treating, storing, and distributing water to users. The second phase—life cycle inventory analysis (LCI)—is related to data collection and calculation procedures that quantify the relevant inputs and outputs after defining the system limits. The life cycle impact assessment (LCIA) phase aims to analyse the significance of potential environmental impacts through the results obtained in LCI [161]. The methods used for LCIA can be classified into two categories according to the approach: midpoint and endpoint. The midpoint methods, which correspond to the intermediate results, cause less uncertainty since they restrict the modelling to relatively early stages and group the results into the midpoint categories. Endpoint methods model the cause-and-effect chain and add high uncertainty [162].

ISO 14040 [161] highlights that LCIA is not a complete assessment of the process or product's environmental issues and only shows the results of the environmental issues defined in the goal and scope. There may be significant differences in LCIA due to the weight given to impact categories and inventory data. Therefore, it is essential to specify the system limits carefully.

Loubet et al. [163] developed an LCA method adapted to the urban water systems analysis called water system life cycle assessment (WaLA). The method reduces the urban water system complexity, ensuring good process representation and fulfilling the LCA requirements. The WaLA model is based on a framework that uses a "generic component" that alternatively represents water technology units and consumers with their associated water flows and the impacts of water deprivation, emissions, operation, and infrastructure.

The WaLA model was implemented in the urban water system of the suburban region of Paris (France) by Loubet et al. [164]. The goal was to verify the model's ability to provide an environmental understanding of issues related to future trends that influence the system (e.g., evolving water demand and increasing water scarcity) or policy responses (e.g., water resources and technologies). For this, the 2012 scenario and several forecast scenarios for 2022 and 2050 were evaluated. The scenarios were designed using the WaLA modelling tool, implemented in Simulink/Matlab (Mathworks, Natick, MA, USA). The life cycle inventories of technologies and user components included water quantity and quality changes, specific operations (electricity and chemicals), and infrastructure data (building materials). The authors concluded that the model could provide information that assists with decision making for future policies.

Hasik et al. [165] performed an LCA of a net-zero energy and net-zero water building's decentralised water system. They compared the results with two other buildings: conventional and water-efficient ones, which used centralised water systems. The net-zero building (NZB) is located in Pittsburgh, Pennsylvania, USA and is an award-winning green construction. The water-efficient building has the same internal features as the NZB but without the advanced on-site treatment system. The results show that although the NZB performed better than the conventional building in most categories, such as in the eutrophication impact, the water-efficient building generally performed better than the NZB. The NZB's lifetime and septic tank aeration were considered essential factors in the NZB's impacts.

Given the fast development of LCA in recent years, Table S1 in the Supplementary Materials shows the goals, functional unit, time of analysis adopted, and the main results of studies that assessed water use systems through LCA.

Regarding the studies that evaluated the environmental impacts of the different stages of the urban water cycle through LCA, Amores et al. [166] found that the distribution step accounts for most of the impacts. The authors emphasised that this result highly depends on the orography and the distance from the abstraction sites to the consumer.

Lemos et al. [167] reported that the abstraction and treatment stage was the most relevant one for almost all the impact categories assessed, mainly due to the higher electricity consumption in these stages.

Among the reviewed LCA studies that compared rainwater-harvesting systems [168–170] or integrated systems (rainwater and greywater reuse) [171–173] with centralised systems, as usual, the majority reported the alternative systems showed better environmental performance than the conventional ones. However, in the work of Chang et al. [172], although the integrated rainwater and greywater system had shown lower greenhouse gas emissions, it presented an energy demand equal to that of the conventional process. Yan et al. [174] assessed the impacts of a device that treats harvested rainwater to meet potable water standards. The potable water produced showed worse performance than the water from a centralised supply due to significant differences in the magnitude of the throughput between a city-scale water treatment unit and a single point-of-use treatment device.

Regarding the scale and the implementation place of water-harvesting systems in LCA studies, the adoption of the neighbourhood scale with collective-use reservoirs was recommended [175]. In addition, implementing water-harvesting systems was mainly encouraged in cities with compact population densities, where environmental impacts are lower than diffuse densities [176].

Rashid et al. [177] compared the environmental impacts of water-harvesting systems according to the tank material: high-density polyethylene (HDPE), low-density polyethylene (LDPE), ferrocement, and steel. The authors found that the HDPE tank had lower impacts in almost all categories considered. One of the LCA studies assessed replacing ordinary taps with water-saving taps on a university campus [178]. With the replacement, there was a reduction of up to 26.2% in all impact categories evaluated.

LCA has also been used to assess sustainable practices in rainwater management, such as infiltration trenches and permeable pavements. Petit-Boix et al. [179] reported that implementing infiltration trenches could potentially avoid environmental impacts caused by floods. According to Vaz et al. [180], with the use of permeable pavements, the energy savings due to the decrease in the amount of treated water by the water utility company are higher than the energy required to keep the system in full operation (energy consumption for pumping water) over the life cycle. However, the system has high embedded energy in the materials adopted, and it is impossible to guarantee a negative energy balance. The balance would be negative when the energy savings due to the decrease in the water treated by the water utility were higher than the sum of the embedded energy, transport energy, and energy used for pumping water. The authors suggested further research evaluating other types of coatings and layers for permeable pavements to make them more sustainable.

As pumps are generally used to supply rainwater for consumption in buildings, the energy use in this step has been studied in the literature. In Australian studies, the energy associated with rainwater pumping ranged from 1.4 to 1.8 kWh/kL, which is higher than the energy needed for a centralised water supply (0.06–1.84 kWh/kL). However, the energy efficiency of pumps can be improved by determining the proper pump size [181,182], with the use of pressure vessels [181] (reducing the energy needs by 30–36% [182]) and header tanks in double-storey housing [181] (energy savings of 58–79% with a 300-L header tank [182]). Moreover, the energy needed for water pumping decreases as the flow rate increases [181,182].

Due to the complexity of the calculations involved and the impact quantification, the use of computer programmes to perform LCA has increased, such as GaBi (Sphera, Chicago, CA, USA), Umberto (Ifu Hamburg, Hamburg, Germany), OpenLCA (GreenDelta, Berlim, Germany), and SimaPro (PRe-Sustainability, Amersfoort, The Netherlands) [150]. Rodrigues et al. [183] stated that there is a remarkable similarity between the computer programmes, and the cost–benefit analysis should be considered when buying one of them.

Brudler et al. [184] quantified the environmental impacts caused by adaptation strategies for climate change. They showed that it is advantageous to carry out environmental assessment of the implementation of rainwater management systems in the initial phases of

the planning process. The same was observed by Schulz et al. [185], who stated that, ideally, detailed assessments of the urban water systems' sustainability should be carried out during the planning phase to inform the decision-making process. However, the method has limitations that must be taken into account. According to Ribeiro [150], the most significant limitation is the incompleteness and lack of reliability associated with the data sources. Furthermore, the large amount of data required to perform the analysis makes it difficult to use LCA [150], resulting in significant time and cost associated with performing detailed environmental analyses [185].

*5.2. Water Balance Modelling*

The impacts of urbanisation on the water balance are known. The surface waterproofing due to the soil's usage and occupation in the urban environment caused an increase in the maximum river flow peaks, increasing the possibility of flooding. Carvalho [186] highlights the main changes caused in the hydrological cycle resulting from urbanisation, such as increasing surface runoff and maximum flow peaks and reducing infiltration rates, groundwater recharge, and evapotranspiration.

Urban water management is urgent and implies some pertinent issues related to potable water supply, wastewater, and rainwater treatment, environmental impact reduction and waterborne diseases, and operational and infrastructure cost mitigation. When combined, such issues represent a challenge for public administration.

Eshtawi et al. [187] demonstrated the importance of using water modelling integrated with the water systems' sustainable urban planning process. Developed in the Gaza Strip urban area, the study provides a complete system view, quantifies the surface and groundwater interactions in detail, exhibits new indices related to the urban area expansion, and creates realistic scenarios inferred from possible decision making.

Locatelli et al. [188] highlighted the importance of investigating long-term changes in water balance due to urbanisation interference and its influence on the groundwater regime. When evaluating the impact in urban areas, where rainwater infiltration into the soil through shallow wells is common, the authors concluded that such a system, combined with urbanisation, affects the entire water balance. Thus, recharge is increased, evapotranspiration is reduced, and the risk of flooding caused by rising groundwater levels is also increased.

More than 140 research studies (from 1990 to 2016), which used computer simulations to assess urban water cycles, were reviewed by Peña-Guzmám et al. [189]. Among such studies, 37 addressed rainwater harvesting, mainly in Australia. According to the authors, water, contaminants, energy, and chemicals are the input water components which integrate into the urban water cycle. Water comes from two primary sources: supply (from the surface and underground) and rainfall. Contaminants correspond to pollutants carried by surface runoff and contained in wastewater. Energy is important due to its environmental effects, such as greenhouse gas emissions and the use of natural resources. Water treatment, supply, and heating systems are closely related to energy. Chemicals used to treat potable water and wastewater have potential environmental and health impacts [189].

Naserisafavi et al. [190] assessed some scenarios of alternative water systems (rainwater and greywater harvesting) in the net-zero water building (NZWB) context. The authors conducted a water balance analysis, life cycle assessment, and economic assessment to define the best scenario among those proposed. The study was conducted by considering a mixed-use building located in Melbourne, Australia. The following scenarios were considered: (1) the building's total water demand would be supplied by main water, (2) rainwater harvesting usage for toilet flushing and irrigation, (3) untreated greywater usage for subsurface irrigation, (4) on-site treated greywater usage for toilet flushing and irrigation, (5) rainwater usage for toilet flushing and greywater for subsurface irrigation, and (6) rainwater usage for toilet flushing and greywater (treated on-site) for irrigation. According to the results, scenario 6 was the most efficient in terms of main water consumption and reduction in surface runoff, with 72% feasibility of on-site greywater treatment.

Regarding greenhouse gas emissions (GHG), scenario 3 was the one with the lowest annual rate. According to the authors, the building analysed may not be an NZWB if it is only supplied by rainwater. Therefore, they concluded that in highly populated buildings with considerable space to irrigate and commercial and residential use (mixed buildings), a single type of alternative supply source will not be sufficient to meet the water demand. If adequate space is available, hybrid systems, such as the mutual use of rainwater and greywater systems, are recommended.

On a city scale, Crosson et al. [95] developed a model to evaluate the necessary network of rainwater-harvesting systems to achieve net-zero urban water in the city of Tucson, Arizona, USA. The authors used a daily water balance model adjusted to achieve the smallest required storage volumes to reach net-zero urban water at 10 years of daily rainfall. Four scenarios were investigated, considering a fully decentralised system (total water demand was met with rainwater) and a decentralised-centralised hybrid system (imported water demand was partially met with rainwater), with 0% and 30% conservation efficiency, respectively. The most financially and physically viable scenario was to replace only the imported water while assuming 30% demand conservation. In this case, the median required storage was $0.282 \text{ m}^3/\text{m}^2$ of the roof. It was shown that net-zero urban water could be achieved with rainwater harvesting with large storage volumes in extreme climates with multi-year droughts.

Comprehending how urbanisation may affect groundwater recharge time and quality is a prerequisite for mitigating water scarcity and identifying contamination vulnerability [191]. Simulation models provide an answer to this question. Generally, the modelling goal is to present water allocation strategies of a complex system in a given period [192].

The water balance study requires complete knowledge of all water resources aspects, including surface and groundwater runoff, their interaction, and water usage. Some models that separate surface and groundwater runoff are developed for particular applications by using methods to simplify hydrological processes. Integrated hydrological models consider both surface and groundwater runoff simultaneously [193]. The results found by water balance modelling contribute to the criteria and solution definitions which may be adopted to improve water flows and reduce impacts on urbanisation. Some examples of water balance modelling applications are shown in Table 7.

Several commercial and free models simulate the water cycle using partial or total combinations of the constituent elements. Researchers, academics, urban water resource administrators, and urban infrastructure designers must know the varied applications of water simulation models. Using such models, designing integrated solutions for the different components of the urban water cycle is possible. The model's usage may help ensure solid economic investments' viability and establish technical arguments for policy and guideline creation oriented towards sustainability [189].

Peña-Guzmán et al. [189] presented the applications of 17 simulation models, Aquacycle (Modelling Toolkit, Monash University, Melbourne, Australia), Urban Volume Quality (CSIRO, Melbourne, Australia), MIKE URBAN (DHI, Hørsholm, Denmark) and Urban Cycle (University of Newcastle, Newcastle, Australia). According to the authors, the most used models are Urban Volume Quality (UVQ–CSIRO, Melbourne, Australia), Aq-uacycle (Modelling Toolkit, Monash University, Melbourne, Australia), and MIKE URBAN (DHI, Hørsholm, Denmark). Such models were applied in more than 50% of experiments. Several models performed rainwater-harvesting applications, but it was more frequent when using Aquacycle.

**Table 7.** Research studies about water balance modelling.

| Author | Main Objective | Main Results |
| --- | --- | --- |
| Willuweit and O'Sullivan [194] | Develop a dynamic water simulation model which associates urban water balance concepts with a land use dynamics model and a climate model. | A model capable of satisfactorily predicting water demand and stormwater runoff. |
| Haase [195] | Analyse the impact of urban land use change on the urban water balance over 130 years in Leipzig, Germany. | Reduction of evapotranspiration and groundwater recharge, in addition to increased direct runoff. |
| Carlson et al. [191] | Investigate the impact on groundwater quality and quantity caused by urbanisation in semi-arid regions of the USA, where the use of artificial recharges is common. | Contribution to groundwater recharge and deterioration of water quality. Need to reduce the contaminants' arrival in recharge areas to protect future groundwater resources. |
| Barron et al. [196] | Assess land use change in the southwestern Australia watershed. | Urban development in the region reduces evaporation and evapotranspiration. It increases infiltration rates (due to direct infiltration from roofs and runoff from roads), generating harvestable water which could improve environmental flows if used for public and private supply. |
| Albertin et al. [192] | Evaluate quantitative and qualitative water availability in the Sapucaí-Mirim River Basin in São Paulo, where the water has multiple uses for domestic and industrial consumption and electricity generation. The simulation model used was MIKE BASIN (DHI, Hørsholm, Denmark). | The need for pollution control and prevention because, although water availability is sufficient to meet the demand, the water quality is being degraded. The leading cause of deterioration of the Sapucaí-Mirim River is the release of untreated domestic sewage. |

Aquacycle is a free model representing the urban water cycle by sequentially simulating the processes of potable water supply, hydrology (precipitation and evapotranspiration), and wastewater on a daily time scale. Constituent elements are permeable and impermeable surfaces, evapotranspiration, water usage for human consumption and irrigation, soil infiltration, centralised systems' losses and leaks, and rainwater harvesting and reuse systems [197]. UVQ is an Aquacycle expansion. The model was developed to rapidly assess the impacts of central conventional and non-conventional urban water supply options, rainwater, and wastewater on the total water cycle [198]. UVQ and Aquacycle are models with grouped parameters that do not require extensive input data, simplifying their use [189].

The first simulation developed using Aquacycle was performed by the programme's authors, Mitchell et al. [197], concluding that the programme was able to satisfactorily simulate the Wooden Valley basin's water balance in Australia. However, it was pointed out that the model should be tested in other watersheds with different climates and topography, land use, and occupation characteristics. Satisfactory simulations in other locations were later confirmed by studies conducted in [121,199], Egypt [200], Israel [201], South Korea [202,203], Greece [204], and others. Lekkas et al. [204] found that the model is generic and may be applied to any urban watershed. Pak et al. [202], using a calibration sensitivity analysis of the Aquacycle model, considered it applicable and useful for an integrated approach to watershed management to investigate the effects caused by water reuse. Table 8 shows the objectives and results of some authors who used Aquacycle.

Despite the management and administration potential of water balance models, Peña-Guzmán et al. [189] concluded that such tool usage was generally focused on academia and not decision-making environments. However, the academics' role in this situation may not be neglected. Creating computer programmes and other approaches allows the creation of a decision-making tool that integrates technical, environmental, economic, and social concepts to visualise different trends or possible scenarios quickly.

**Table 8.** Research studies about water balance modelling using Aquacycle.

| Author | Main Objective | Main Results |
|---|---|---|
| Lee et al. [203] | Study the effects of land use change and water reuse options in an urban water cycle in the Goonja drainage basin in the metropolitan region of Seoul (South Korea). | The chronological effects of urbanisation were evaluated from 1975 to 2005, and the proportion of impervious areas ranged from 43% to 84%. Urbanisation generated a severe water cycle distortion: it caused reductions in evapotranspiration (29%) and groundwater recharge (74%), in addition to an increase in surface runoff (41%). The authors concluded that wastewater reuse is more advantageous than rainwater usage, as it provides a consistent water supply throughout the year to Korea, where the rainfall distribution is very variable and concentrated during the summer. |
| Sharma et al. [199] | Evaluate scenarios for establishing future policies for water services in Canberra (Australia) using water balance modelling. | Water balance and quality analyses were performed using the Aquacycle and Music models, respectively. The peak flow and flow reduction analyses were conducted using the Purrs model. Potable water savings were more significant with demand management tools or the combination of greywater and rainwater usage. Analysing rainwater and greywater uses separately, there was more significant potential for potable water savings by rainwater harvesting than by greywater reuse, with the additional benefit of reducing the peak flow of surface runoff promoted by the harvesting systems. |
| Donia et al. [200] | Develop a model to represent Alexandria's urban water system based on the Aquacycle computer programme. | Results demonstrated a potential possibility for on-site recycling and reuse, which will need to be evaluated in detail regarding costs and environmental impacts. |
| Duong et al. [201] | Evaluate water balance modelling and some energy aspects of the implementation of urban water management strategies in Tel Aviv, Israel. | Strategies' effect on the total imported water amount into the city was a reduction of 10% with rainwater harvesting and 32% with wastewater reuse. |

## 6. Discussion

Demand management is essential to control water consumption in buildings and ensure the sustainable management of potable water. In this context, installing water-saving appliances and increasing user awareness are relevant instruments for reducing consumption. However, even before the proposal of measures to save potable water, the use of smart measurement tools has been notable and decisive for achieving higher control of leaks and determining consumption profiles and water end uses, classifying them as potable and non-potable. These systems capture information on water use and transmit it in real time, allowing consumption control. Among other benefits, understanding the meters' consumption pattern encourages consumers to take conscientious measures regarding water conservation.

The assessment of the potential for potable water savings from using alternative sources, such as rainwater for non-potable purposes in buildings, is possible by analysing the water end uses. As for the data on consumption profiles, a high difference was observed between the water end uses due to climate, socioeconomic class, household appliances, the use of energy-saving equipment, culture, etc. However, despite the differences, the potential for replacing potable water with lower quality water for non-potable uses is significant. The water consumption for toilets represents about 20% of the total water consumption in residential buildings, 44% in public buildings, and 68% in commercial buildings. In this sense, alternative sources play an important role in sustainable water supply management if the systems are sized to meet the demand. Although computer programmes allow the simulation of rainwater-harvesting systems with different values for rainwater demand,

the closer these values are to reality, the more accurate the environmental assessment of systems in the hydrological cycle.

Regarding rainwater-harvesting system design, it is relevant to consider climate change, as rainfall data are essential for calculating storage tank capacity. Geraldi and Ghisi [105] showed that 10-year series of daily rainfall data might be sufficient to size rainwater tanks. In this context, it is suggested that further research assess not only the impact of the size of the data series but also the impact generated by climate change on the design of rainwater-harvesting systems. It is also essential to highlight the importance of further research evaluating the impact of design variables on the performance of rainwater-harvesting systems. In these studies, it is necessary to consider predicted rainfall data series without neglecting climate change.

It is also important to highlight that the use of community rainwater reservoirs integrated with the centralised system can be a viable alternative in terms of the sustainable development of urban water systems at the cluster scale. Among the advantages, according to Sharma et al. [205], these systems can increase the resilience of urban water systems to the impacts of climate change and reduce the impact of urban development on the natural environment. This type of practice is not yet common in Brazilian cities and lacks national research on its technical and financial feasibility, environmental impact, and social acceptance. Still, when it comes to community reservoirs, it would also be interesting to develop research for the use of flood containment tanks, such as rainwater-harvesting reservoirs, through intelligent systems.

Rainwater harvesting promotes a more rational water usage and improves the urban management of this resource. The studies reviewed herein point out that using rainwater can significantly reduce the water flow volumes and peaks, avoiding flooding, primarily when associated with green roofs. However, it is essential that analyses of the impact of rainwater-harvesting systems on drainage systems also consider predicted climate data, as climate change is associated with extreme rainfall events [206].

Additionally, reducing rainwater contamination risks is essential, since contaminated water can cause risks to the user's health. For non-potable purposes, the system must have first-flush devices and filters. Furthermore, periodic cleaning of storage tanks and catchment areas is also recommended. In order to use rainwater for potable purposes, aside from the cautions already mentioned, at minimum, disinfection must be carried out. In addition, since pesticides and heavy metals can contaminate the rainwater, it is recommended to analyse the quality parameters and conduct toxicological tests to ensure safe consumption without damaging health in the short, medium, or long term.

Another influence on water quality studied by researchers is the type of roof. It was observed that, despite the great importance of the use of green roofs for the microclimate and for the reduction of drainage peaks and acid rains in the urban environment, when it comes to the use of rainwater (conventional green roofs), some quality parameters could be a little compromised, especially for the turbidity parameter. The use of unconventional green roofs that also serve for water treatment, similar to what is performed in wetlands, is the topic for other studies involving green roofs and rainwater harvesting, such as that by Xu et al. [147].

Interest in the zero-water building (ZWB) topic has grown, though real cases of ZWBs are rarely found. According to Asadi et al. [94], this lack of cases is because ZWBs include the consumption of potable and non-potable water, alternative sources of water on site, a fresh water supply, alternative water supplementation, and water returned to the original water resource. There are also benefits of harvesting rainwater in net-zero water buildings. Considering that there are water losses in the treatment of effluents and that it would be necessary to use very large rainwater reservoirs to meet the demand in net-zero water buildings, the concomitant use of rainwater harvesting and effluent reuse becomes feasible. However, it is always important to assess the environmental impact of these systems so that the best choice is made.

Considering the benefits of the water use systems presented, a more comprehensive environmental assessment of the impacts that the installation of these systems can cause on the environment and the water cycle is still necessary. LCA allows quantifying the impacts generated of a system or product from acquiring raw materials to production, use, recycling, and final disposal. In this way, it is possible to propose improvements in specific life cycle phases and compare the generation of impacts of different water supply systems. It was observed that the contribution of each life cycle phase to environmental impacts varies between studies and is dependent on the type of system, materials, and energy sources used, as well as issues of relief and local distances. Thus, LCA studies often show specific situations, and their results should not be generalised. Despite the differences between the results, the rainwater-harvesting systems or integrated systems (rainwater and greywater reuse) showed environmental impacts lower than conventional centralised systems in most of the studies on LCA.

Modelling the water balance in turn provides information on the impact of the urban water cycle system, allowing the comprehension of how urbanisation may affect water quality, groundwater recharge time, and surface runoff. Therefore, the environmental water systems' impact analysis must consider the effects caused by the materials, chemicals, and energy used during the systems' lives, as well as the quantitative and qualitative impacts caused in the water cycle. Thus, life cycle analysis and water balance modelling methods are complementary in this process. When systems are not designed from a holistic perspective, unexpected infrastructure costs or deleterious environmental effects may be generated, such as greenhouse gas emissions or high energy consumption [189].

## 7. Conclusions

Integrated demand management proved to be a great solution for reducing urban consumption of potable water. In this context, smart measurement systems are the ones that have shown the best results. Although these systems still have a high cost of implementation, they promote environmental, social, and scientific benefits. The use of smart measurement systems can reduce water losses in leakage due to rapid diagnosis, encourage conscious consumption, and help research by defining consumption profiles and water end uses.

Due to several factors influencing consumption profiles, for case studies, the water consumption diagnosis must be made in the building under study, especially in commercial and public buildings with a higher number of influences. On the other hand, the economic factor's influence on water consumption is remarkable. Furthermore, it is easy to identify low-, medium-, and high-standard areas in cities. Thus, the economic factor can be used in large-scale research, which requires mapping urban buildings.

The definition of water end uses in buildings showed great potential for potable water savings by using alternative sources (such as rainwater harvesting) for non-potable purposes, allowing the reestablishment of water availability over time. Based on the studies on the use of rainwater in buildings, it was observed that the potential for potable water savings was satisfactory (generally between 20% and 65%). The quality of the rainwater collected was considered appropriate for non-potable uses, and the social acceptability of rainwater-harvesting systems was shown to be high. In addition, the literature reported that implementing rainwater-harvesting systems can reduce surface runoff by 13–91%.

The analysis of the potential for rainwater use and the potable water savings due to rainwater use does not consider other possible benefits and impacts of the systems on the water flow and the environment. Therefore, there is a need for a holistic assessment of rainwater-harvesting systems considering the environmental impacts generated by such techniques. In this context, LCA and water balance modelling tools have been used in the literature, focusing on urban water management. For example, the environmental impact generated by rainwater-harvesting systems and integrated rainwater systems and greywater reuse throughout its life cycle has been evaluated through LCA studies. Most studies reported that such alternative systems showed better environmental performance

than centralised systems. Based on studies related to the water balance method, it is possible to assess the rainwater-harvesting systems' contribution to flood reduction. Studies based on the water balance provide a broader and more complete view of urbanisation's impact on a given region's water situation.

Usually, LCA and water balance modelling tools are used separately both for studies of the urban scope and for studies of single buildings. However, the study of rainwater-harvesting systems integrated into the urban environment using both tools is essential, considering their impact on the water balance and the consumption of materials and energy.

**Supplementary Materials:** The following supporting information can be downloaded at: https://www.mdpi.com/article/10.3390/w14172716/s1, Table S1: Research studies about LCA of water use systems.

**Author Contributions:** Conceptualisation, A.T. and E.G.; methodology, A.T.; investigation, A.T., T.P.S. and J.K.M.; writing—original draft preparation, A.T.; writing—review and editing, E.G., T.P.S. and J.K.M.; visualisation, T.P.S. and J.K.M.; supervision, E.G. All authors have read and agreed to the published version of the manuscript.

**Funding:** This research received no external funding.

**Data Availability Statement:** Not applicable.

**Conflicts of Interest:** The authors declare no conflict of interest.

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
