# Peer review of "Comprehensive Environmental Assessment of Rainwater Harvesting Systems: A Literature Review"

_water, doi:10.3390/w14172716_

Round 1
Reviewer 1 Report
Review comments on the manuscript “Comprehensive environmental assessment of rainwater harvesting systems: A literature review”
Authors have developed the manuscript based on literature review, which is mainly focused on publications from Brazil. Some other publications from other part of the globe are suggested for authors to consider to further add value to their manuscript:
Reviews of rainwater tanks based on literature have also been been provided in the following references. Authors are suggested to check these publications for any new information to include:
Sharma, A.K., Cook, S., Gardner, T., and Tjandraatmadja, G. (2016) Rainwater tanks in modern cities: A review of current practices and research, Journal of Water and Climate change, 7(3),445-466. doi: 10.2166/wcc.2016.039
Sharma, A.K., Gardner, T. (2020) Comprehensive Assessment Methodology for Urban Residential Rainwater Tank Implementation, Water 2020, 12, 315; doi:10.3390/w12020315
Line 9: Suggested to change “based only on economic analysis” to “ based on primarily on economic analysis”
Line 21: Life cycle assessment and water balance methods are indicated essential to evaluate rainwater tanks. However, life cycle costing is another important factor for consideration along with water balance for economic assessment.
Please see reference below for more information:
Murray R. Hall, Thulo Ram Gurung and Kym Whiteoak (2015) Economics of individual and communal rainwater tank systems in Rainwater Tank Systems for Urban Water Supply Design, Yield, Energy, Health Risks, Economics and Social Perceptions; Edited by Ashok K. Sharma, Donald Begbie and Ted Gardner; IWA Publication; ISBN: 9781780405360
https://iwaponline.com/ebooks/book/252/Rainwater-Tank-Systems-for-Urban-Water-Supply
Line 76: The rainwater analysis has also been conducted to consider the possiblity of net zero water use goal in the following study using water balance, life cycle cost (LCC) and multi-criteia decision making:
Naserisafavi, N., Yaghoubi, E., Sharma, A. K.(2021) Alternative water supply systems to achieve the net zero water use goal in high-density mixed-use buildings Sustainable Cities and Society https://doi.org/10.1016/j.scs.2021.103414
Lines 142, 150 and section 4.1: Monitoring water usage, using smart meters, can also help managers to check state policies if the rainwater tanks achieve objectives as planned through state policies. Please refer the following publication:
Umapathi, S., Chong, M., Sharma A.K. (2013) Evaluation of plumbed rainwater tanks in households for sustainable water resource management: A real-time monitoring study, Journal of Cleaner production, 42(2013), 204-214. doi.org/10.1016/j.jclepro.2012.11.006
Line 378: The influence of different variables in rainwater tank modelling (roof area, tank volume, demand, initial loss and continuing loss factor) has been described in detail in the following publication:
Alison M. Vieritz, Luis E. Neumann and Stephen Cook (2015) Rainwater tank modelling in Rainwater Tank Systems for Urban Water Supply Design, Yield, Energy, Health Risks, Economics and Social Perceptions; Edited by Ashok K. Sharma, Donald Begbie and Ted Gardner; IWA Publication; ISBN: 9781780405360
https://iwaponline.com/ebooks/book/252/Rainwater-Tank-Systems-for-Urban-Water-Supply
Line 487 Section 4.2 – Rainwater tanks also have impact on the peak factor reduction in water supply systems. Please see the following publication:
Lucas, S. A., Coombes, P.J. and Sharma A.K. (2010) “The impact of diurnal water reuse patterns, demand management and rainwater tanks on the supply network design” Journal of Water Supply and Technology-Water Supply-WSTWS 10(1), 69-80. https://doi.org/10.2166/ws.2010.840
Line 610 Section 5.1 : Life Cycle Assessment also needs information on the use of energy in rainwater supply through tank as pumps are generally used. The following researchers have conducted study for the estimation of energy usage in rainwater tank supply:
Grace Tjandraatmadja, Monique Retamal, Shivanita Umapathi and Guenter Hauber-Davidson (2015) Understanding energy usage in rainwater tank systems through laboratory and household monitoring in Rainwater Tank Systems for Urban Water Supply Design, Yield, Energy, Health Risks, Economics and Social Perceptions; Edited by Ashok K. Sharma, Donald Begbie and Ted Gardner; IWA Publication; ISBN: 9781780405360
https://iwaponline.com/ebooks/book/252/Rainwater-Tank-Systems-for-Urban-Water-Supply
Tjandraatmadja, G., Pollard, C., Sharma, A.K., Gardner, T (2013) How supply system design can reduce the energy footprint of rainwater supply in urban areas in Australia, Water Science and Technology: Water Supply, IWA Publishing 06/2013; 13(3):753-760. doi: 10.2166/ws.2013.057
Moreover, the manuscript is mainly focused on individual rainwater tanks, however communal rainwater tanks can also be another option in cluster scale develoments. It is suggested that authors should consider including some aspects of communal rainwater tanks. The following references are suggested for consideration:
Stephen Cook, Ashok K. Sharma, Thulo Ram Gurung, Luis E. Neumann, Magnus Moglia and Priya Chacko (2015) Cluster-scale rainwater harvesting in Rainwater Tank Systems for Urban Water Supply Design, Yield, Energy, Health Risks, Economics and Social Perceptions; Edited by Ashok K. Sharma, Donald Begbie and Ted Gardner; IWA Publication; ISBN: 9781780405360
Cook, S., Sharma, A., Chong, M. (2013) Performance Analysis of a Communal Residential Rainwater System: a case study in Brisbane, Australia. Water Resources Management, 27, 4865-4876, DOI 10.1007/s11269-013-0443-8
Sharma, A., Cook, S., Chong, M. (2013) Monitoring and validation of decentralised water and wastewater systems for increased uptake. Water Science and Technology, 65(11), 2576-2581, IWA Publication doi: 10.2166/wst.2013.168
Conclusion: Authors may like to add the application of Life Cycle Costing in the overall assessment of rainwater tanks alongwith LCA and water balance analysis.
Reviewer 2 Report
Dear authors,
I congratulate you for this extensive work. Here are my comments for improving this manuscript.
1. Restructure your manuscript. In the current form a lot of unnecessary details are presented. There is no clear flow of the information in any section.
2. Where is methodology??? The authors are suggested to use PRISMA protocol of systematic review.
3. There is no need for discussion section. It is a review paper and every section is a discussion.
4. Check for language corrections
Reviewer 3 Report
The article is not very conclusive, but it contains unpublished contents of scientific interest. The main comment refers to the References. On the one hand, some references are very marginal to the topic, but, on the other hand, there is a lack of references in some recent trends in the context of rainwater harvesting in buildings. Authors are recommended to include references, albeit brief, on the integration of green roofs with rainwater harvesting systems and on the contribution of rainwater harvesting to nearly zero water buildings, for example.
Round 2
Reviewer 1 Report
Well done
Author Response
The authors are grateful for your revision!
Reviewer 2 Report
The authors have justified their stance which I respect. In present form I recommend the publication of this manuscript
Author Response

(The authors gave the same response as above.)

Reviewer 3 Report
The authors sought to answer the questions raised, related to the omission of references in relation to the role of rainwater harvesting in buildings with the development of nearly zero water buildings and in relation to the integration of green roofs with rainwater harvesting systems, but the references indicated are relatively marginal. It is suggested to the authors a more elaborate research and the selection of more focused references.
